# Knowledge, attitudes, and practices on anthrax in selected game management areas in Zambia

Exillia Kabbudula[1], Laila Gondwe[1], Chitwambi Makungu[2], Mtui-Malamsha N. Jesse[2], Kezzy Besa[1], Suwilanji S. Sichone[3], Noanga Mebelo[1], Mwila Kayula[2], Geoffrey Mainda[2], Fredrick M. Kivaria[4], Charles Bebay[5], Baba Soumare[4], Suze P. Filippini[2], Chisoni Mumba 📶[1]*

1 Department of Disease Control, School of Veterinary Medicine, University of Zambia, Lusaka, Zambia, 2 Emergency Centre for Transboundary Animal Diseases (ECTAD), The Food and Agriculture Organization of the United Nations (FAO), Lusaka, Zambia, 3 Pangolin Protection, Wild Crime Prevention (WCP), Chilanga, Zambia, 4 Emergency Centre for Transboundary Animal Diseases (ECTAD), The Food and Agriculture Organization of the United Nations (FAO), Nairobi, Kenya, 5 Emergency Centre for Transboundary Animal Diseases (ECTAD), The Food and Agriculture Organization of the United Nations (FAO), Rome, Italy

* sulemumba@yahoo.com, cmumba@unza.zm

## Abstract

In this study, we investigated anthrax, a zoonotic disease, at the human-wildlife-livestock interface in Zambia, focusing on Simalaha Conservancy, Blue Lagoon, and Lochinvar National Parks. These areas represent key points where illegal wildlife trade and anthrax risk coexist. Although anthrax remains endemic in Zambia, there is limited data on community knowledge, attitudes, and practices (KAPs) related to anthrax transmission and the role of wildlife trade value chains in its spread and maintenance. Therefore, we examined how these community KAPs influence anthrax transmission through the illegal game meat trade. We conducted a cross-sectional study involving 1,187 participants and analyzed data using descriptive statistics and logistic regression. Our findings revealed significant variations between knowledge of anthrax and behavior. While more than 80% of respondents demonstrated positive attitudes and safe practices, their knowledge of anthrax remained limited. Specifically, 72.5% of participants had low awareness of the disease, yet 82.1% acknowledged its severity, and 59.4% recognized vaccination as a control measure. Despite these positive attitudes, misconceptions about anthrax transmission and treatment were widespread. Some community members relied on spiritual or herbal remedies rather than seeking medical intervention. We also identified several risky behaviors that contribute to anthrax transmission. Notably, 14.9% of respondents admitted to consuming meat from animals that had died suddenly, while 46.3% reported handling potentially infected carcasses or animal products without taking safety precautions. We found strong correlations between knowledge, attitudes, practices, and education levels. Our results showed that prior exposure to anthrax information was the strongest predictor of knowledge (β > 1). Our findings highlight the urgent need for targeted public health

**Data availability statement:** All relevant data are within the manuscript and its Supporting information files.

**Funding:** The author(s) received no specific funding for this work.

**Competing interests:** The authors have declared that no competing interests exist.

interventions, improved veterinary services, and stricter enforcement of wildlife trade regulations to reduce the risk of anthrax transmission in these communities.

## Author summary

We investigated the knowledge, attitudes, and practices (KAP) related to anthrax in communities surrounding key wildlife areas in Zambia, where anthrax and illegal wildlife trade both occur. The research revealed that while most respondents showed positive attitudes and safe practices, there were significant gaps in their knowledge of anthrax. Misconceptions about anthrax transmission and treatment were common, and risky behaviors such as consuming meat from animals that had died suddenly were reported. We emphasize the need for targeted public health interventions and stricter wildlife trade regulations to mitigate the risk of anthrax transmission in these communities.

## 1. Introduction

Anthrax, a neglected zoonotic disease, is of economic and public health significance, yet its global distribution remains poorly characterized [1–4]. It is caused by *Bacillus anthracis*, a gram-positive, motile, spore-forming bacterium that mainly affects grazing herbivores [5–7]. Anthrax is endemic, particularly in Western and Eastern provinces of Zambia [8,9].

The human-wildlife-livestock interface plays a crucial role in the maintenance and spread of anthrax [10]. Ecological dynamics at this interface contribute to human mortality from anthrax [11]. The enzootic cycle involves long-term spore persistence in soil and lethal transmission to animals especially herbivores [5,12,13]. In sub-Saharan Africa, bushmeat trade activities contribute to anthrax perpetuation [14]. Wildlife trade brings wild animals in close proximity to humans and provides an interface for zoonotic spillover [15,16]. Vaccination of free-ranging wildlife remains impractical, allowing anthrax to persist in natural ecosystems [13]. Beyond conservation threats, illegal wildlife trade facilitates pathogen exchange through direct human-animal interactions and concealment of infected meat and meat products [17–19]. Illegal value chains for wildlife products often coexist alongside legal value chains and are resilient to the presence of a legal market [20–22], complicating control efforts.

Anthrax is a significant disease affecting various wildlife species, particularly those commonly hunted for food. Large-scale outbreaks have been reported in areas with alkaline soils [13,23], with species like deer, bison, and antelope being highly susceptible [5,13,14]. In North America, a 1962 outbreak in Hook Lake led to 32 bison mortalities from a population of 1,300 [14,24,25], while earlier cases in 1952 resulted in human cutaneous anthrax from exposure to infected carcasses [14]. In southern Africa, at least 52 wildlife species have been affected, underscoring the risks posed by anthrax in ecosystems where wildlife is part of the food chain [14]. The disease

is endemic, particularly in Western, Eastern and some parts of North-Western provinces of Zambia [9,10], where alkaline soils favor the persistence of anthrax spores, contributing to recurring outbreaks. Additionally, underreporting remains a challenge due to limited veterinary extension services, a shortage of trained personnel, and socio-behavioral factors [9,26,27]. Studies show that animals such as kudus, cattle, hippopotamus, giraffe, buffalo, kudu, elephant, puku, wild dog, waterbuck, impala, wildebeest, and hyena are reported to have been infected by anthrax in Zambia [4,28–30].

Between 1990 and 2023, multiple anthrax outbreaks in Zambia affected humans and livestock, primarily originating from wildlife. In 2023, an outbreak in Sinazongwe and Lumezi, linked to infected hippos, resulted in four human deaths [4,28–30]. In June 2023, 26 people developed cutaneous anthrax after consuming contaminated hippo meat, while cattle and goats also died [30]. The 2011 South Luangwa outbreak saw 85 hippo deaths and 521 suspected human cases, with a 1.2% case fatality rate [6,28]. The highest incidence of human anthrax cases occurred in 1990 (220 cases), followed by 248 cases in 1991 and 1998, with fatality rates ranging from 7.2% to 19.1% [29,30]. Between 1999 and 2007, 4.6% of 1,790 cases resulted in death [29,30]. In Western Province, the cattle-to-human infection ratio was approximately 1 human case per 7.4 cattle cases [30] in 1990. The 2016–2017 outbreaks affected both humans and animals in Western and Eastern provinces [2,26], with possible underreporting [2].

Chronic and acute food insecurity drives the consumption of unsafe foods, increasing vulnerability to zoonotic infections [28]. Despite its public health significance, weak surveillance systems underestimate anthrax morbidity, mortality, and socioeconomic impact[30]. Global annual human anthrax cases range from 20,000–100,000. In Zambia, anthrax is endemic and increasing, as evidenced by 2023 outbreaks in Southern Province. This trend threatens public health and exacerbates poverty [3]. The loss of livestock undermines food and nutritional security, as livestock products are essential protein sources [23]. Anthrax's multi-species infectivity and environmental persistence amplify these risks [5]. Weak public health systems and limited surveillance heighten the risk of anthrax outbreaks.

Therefore, this study aimed to assess the knowledge, attitudes, and practices (KAP) on anthrax in communities surrounding Simalaha Conservancy, Lochinvar, and Blue Lagoon National Parks' Game Management Areas (GMAs) in Zambia.

## 2. Materials and methods

### 2.1. Ethics statement

We obtained ethical approval from the Ethics Research Committee (ERES Converge) under reference number 2024-Aug-015. We implemented strict measures to ensure participant confidentiality, avoiding the collection or publication of any identifiable personal information. No data was shared with law enforcement or other authorities, safeguarding respondents' privacy. Additionally, we respected local customs and cultural norms while maintaining data reliability and research integrity. Before data collection, we obtained informed consent from all participants. We clearly explained the purpose of the study, potential risks, and their rights, including the option to withdraw at any time without any consequences. This process ensured voluntary and informed participation while upholding ethical research standards. We obtained verbal consent before administering questionnaires.

### 2.2. Study design

We employed a community-based cross-sectional study design utilizing structured questionnaires to collect quantitative data. This approach enabled the assessment of knowledge, attitudes, and practices (KAP) related to anthrax among residents in selected Game Management Areas (GMAs) of Zambia.

### 2.3. Description of the study sites

We conducted the study in three key locations: Simalaha Community Conservancy, Lochinvar National Park, and Blue Lagoon National Park as shown in Fig 1. Lochinvar and Blue Lagoon National Parks were selected because of the recent

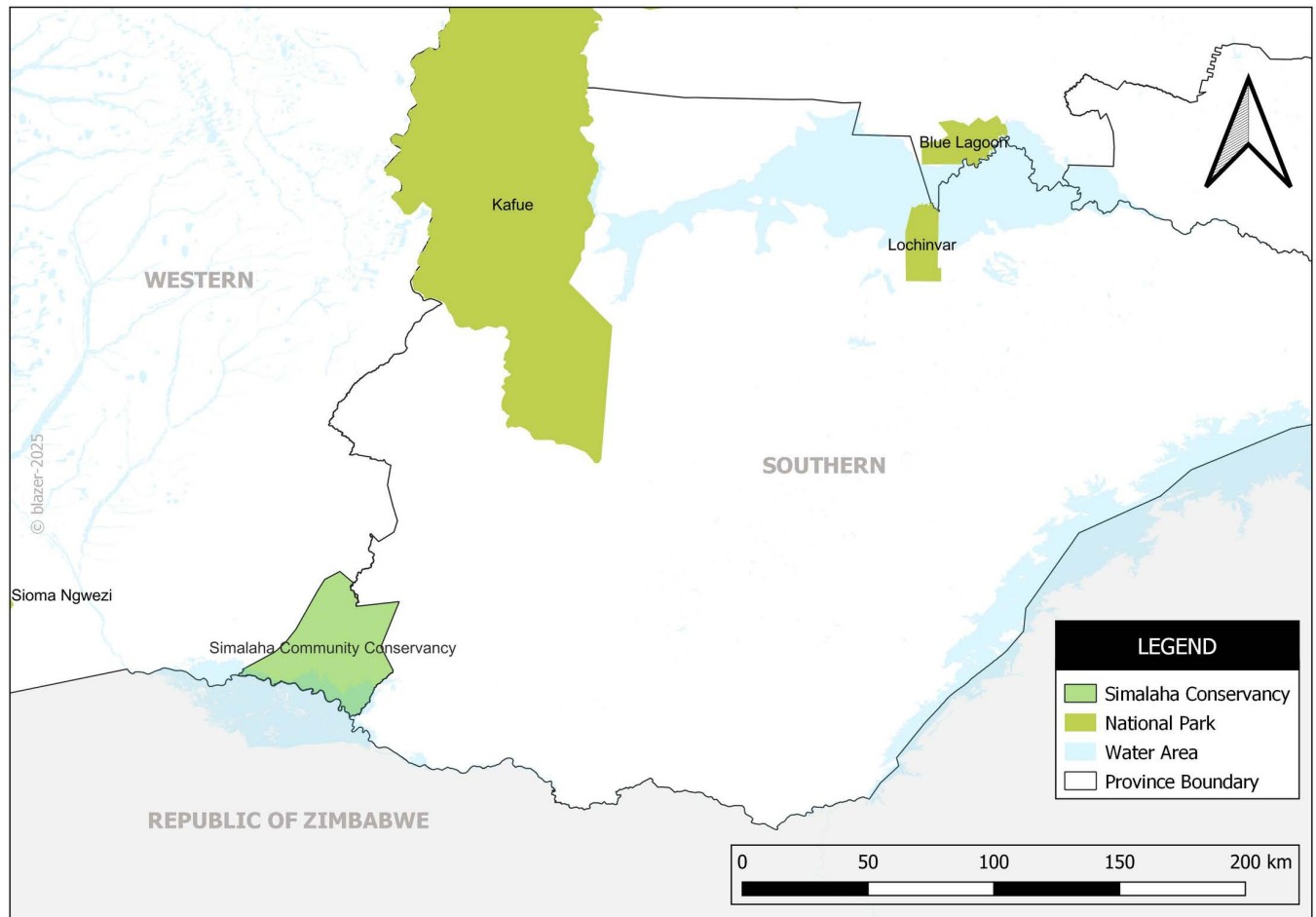

**Fig 1. Map showing the study sites.** Developed by authors. Reprinted from [Zambia_Mosaic_250Karc1950_ddecw] under a CC BY licence, with permission from the Surveyor General, Government Republic of Zambia, original copyright, [1974].

anthrax outbreaks in Southern Province which originated from wildlife (hippos) [31]. Simalaha Community Conservancy was selected because of the endemicity of anthrax disease in Western Province. These study areas also host the largest livestock population in Zambia, a critical human-wildlife-livestock interface area which makes understanding of disease dynamics better [31].

Simalaha Conservancy, located in Zambia's Western Province, spans communities from Mwandi and Sesheke districts. As part of the Kavango-Zambezi Transfrontier Conservation Area (KAZA TFCA), Simalaha serves as a crucial wildlife dispersal corridor connecting Chobe National Park in Botswana with Kafue National Park in Zambia. This region is notable for its community-driven conservation model, where local communities actively participate in managing natural resources. However, its proximity to wildlife also increases the risk of zoonotic disease transmission, including anthrax, due to frequent human-wildlife-livestock interactions [32].

Lochinvar and Blue Lagoon National Parks, situated within the Kafue Flats, were also selected due to their ecological significance and the endemicity of anthrax in these areas. Lochinvar National Park, positioned on the southern edge of the Kafue Flats, features extensive floodplains that support diverse wildlife, including the endemic Kafue lechwe and various bird species. The park's ecological characteristics, particularly seasonal water fluctuations and nutrient-rich soils, create

conditions conducive to anthrax outbreaks. The park covers approximately 428 km² and is classified as a Ramsar wetland of international importance [23]. Similarly, Blue Lagoon National Park, located west of Lusaka, provides critical habitats for waterbirds and herbivores while sustaining local livelihoods through tourism and ecosystem services. The proximity of both parks to human settlements heightens the risk of disease transmission at the human-wildlife-livestock interface, making them relevant study sites for understanding anthrax dynamics.

## 2.4. Sample size estimation

We lacked precise information on the total number of households in each village. Therefore, we used the sample size estimation formula for unknown populations in Epitools (http://epitools.ausvet.com.au/). Assuming unknown household population in study locations, a confidence level of 95%, estimated proportion of 50%, and desired precision of 5%, the necessary sample size was calculated at 385 respondents, assuming random sampling. Since there were 3 study sites, we multiplied 385 by 3 to get a sample size of 1,155 households.

We distributed an equal number of questionnaires across the three study districts. However, we allocated slightly more questionnaires to Simalaha Conservancy in Southern Province, as it extends into Mwandi District of Western Province, an area endemic to anthrax.

## 2.5. Data collection technique

We collaborated with diverse team of experts from three organizations, the University of Zambia (UNZA), Wild Crime Prevention (WCP), and the Food and Agriculture Organization (FAO), to conceptualize and develop data collection tools. We conducted three online meetings to develop a structured questionnaire and plan data collection across three Game Management Areas. The questionnaire was initially drafted in English and circulated among all authors for feedback. This served as a pretest to evaluate its clarity, strengths, and weaknesses, as well as its ability to capture the intended responses. After incorporating the authors' input, we exported the questionnaire into ZOHO software (https://survey.zoho.com/survey) and shared the link with the 26 recruited data collectors (veterinary assistants, agricultural extension officers, and game scouts) through a social media group. This provided an additional layer of pretesting to ensure data quality.

We refined the online questionnaire based on feedback from data collectors, who shared insights on how farmers might interpret the questions. Subsequently, we organized three one-day workshops in each study location to train the data collectors. During these workshops, we observed each data collector administer at least five questionnaires using both online and offline versions on electronic devices.

We piloted the questionnaire in Monze district, which revealed redundancies and areas requiring clarification. Following this pilot, we revised the questionnaire to eliminate repetitive and ambiguous questions. Once finalized, we initiated data collection across all study locations, incorporating input from researchers, farmers, and enumerators. During face-to-face interviews, the questionnaire was translated into the relevant local languages - Tonga, Ila, and Lozi - to ensure effective communication and accurate data collection.

Participants were recruited using community mobilization by local leaders and extension officers. Each selected village was visited, and households were randomly approached until the target number was reached. Enumerators explained the study, obtained verbal consent, and administered the electronic structured questionnaire

## 2.6. Data management and analysis

We systematically stored and managed quantitative data using Zoho Survey, which enabled automated data storage and retrieval. Our research team reviewed the dashboard daily, retaining only fully completed questionnaires.

Using Zoho Survey, we generated descriptive statistics before cleaning, standardizing, and coding the data in Microsoft Excel. We then exported the dataset to Statistical Package for Social Sciences (IBM SPSS Statistics version 25) for

further analysis. We assessed Knowledge, Attitudes, and Practices (KAP) using standardized scoring methods to ensure consistency and reliability.

To evaluate the reliability and internal consistency of our survey instrument, we applied the Kuder-Richardson test (KR-20).

Knowledge was measured using a binary scoring system, classifying responses as correct (1 point) or incorrect (0 points), and categorizing knowledge levels based on Bloom's taxonomy (≥60%, Knowledgeable or <60% Not knowledgeable), after calculating each respondent's percentage score. This approach provided an objective and structured evaluation of participants' understanding.

To assess attitudes and practices, we used Likert scales, allowing respondents to express varying degrees of agreement or frequency. Responses were then grouped into quartiles to identify patterns and differences across participant groups. Unlike simple yes/no responses, this method enabled a more nuanced and meaningful analysis of trends in attitudes, and behaviors. Such trends (positive, neutral, or negative) were further collapsed into binary categories such as good or poor attitudes, and safe or risky practices. Respondents falling within the first quartile (those consistently selecting negative options) were categorized into the poor attitude or risky practice groups. These analytical techniques were selected for their ability to generate clear, comparable, and statistically valid insights, ensuring a comprehensive understanding of the factors influencing anthrax risk at the human–wildlife–livestock interface.

For an in-depth analysis, we first assessed whether key sociodemographic characteristics such as age, gender, education level, and household size varied significantly across the study sites. Pearson's Chi-square test was applied for categorical variables, while Kruskal–Wallis tests were used for continuous or ordinal variables. These tests helped identify statistically significant differences across locations and guided subsequent disaggregation of results.

We then explored correlations (p = 0.05, CL = 95%) between KAP scores and key demographic and behavioral variables to identify potential predictors prior to multivariable modelling. A correlation matrix (Spearman's rho) was generated to examine pairwise relationships between variables, including age, gender, education level, and KAP scores. The study also examined behavioral risk patterns by analyzing self-reported actions such as taking precautions when handling raw meat, consuming meat from unknown sources, and involvement in the informal meat trade. These analyses provided deeper insight into behaviors associated with anthrax risk.

Finally, we used binary logistic regression to examine associations between selected independent variables such as gender, age, and education level, and binary outcomes including knowledgeable versus not knowledgeable. Model fit was assessed using the Hosmer–Lemeshow goodness-of-fit test (P = 0.05), and results were reported as odds ratios (CL = 95%)

## 3. Results

We initially targeted 1,155 participants based on sample size calculations; however, we received 1,193 responses and retained 1,187 after removing incomplete questionnaires.

### 3.1. Socio-demographic characteristics and their association against KAPS

We analyzed demographic characteristics of 1,187 respondents to determine their associations with Knowledge, Attitudes, and Practices (KAP) concerning anthrax. The Kruskal–Wallis H test was used to assess statistical differences in KAP scores across various socio-demographic groups as shown in Table 1.

The majority of respondents (15.3%) were between the ages of 36 and 40 years, followed closely by those aged 21–25 (13.6%), 31–35 (13.3%), and 26–30 (11.9%) years. The results revealed a statistically significant association between age and KAP scores (H = 11.760, *p* = 0.008), indicating that age influences respondents' knowledge, attitudes, and practices regarding anthrax.

Males accounted for 57.7% of the sample, while females made up 42.3%. There was no statistically significant difference in KAP scores between males and females (H = 4.959, *p* = 0.175), suggesting that sex did not influence respondents' anthrax-related knowledge or behavior.

**Table 1. Socio-demographic Characteristics and their association against KAPS.**

| | | N | N (%) | Test statistic (H) | P Value |
|---|---|---|---|---|---|
| Age of Respondent | Below 20 years | 52 | 4.4% | 11.760 | 0.008 |
| | 21-25 Years | 161 | 13.6% | | |
| | 26-30 Years | 141 | 11.9% | | |
| | 31-35 Years | 158 | 13.3% | | |
| | 36-40 Years | 182 | 15.3% | | |
| | 41-45 Years | 138 | 11.6% | | |
| | 46-50 Years | 115 | 9.7% | | |
| | 51-55 Years | 88 | 7.4% | | |
| | 56-60 Years | 63 | 5.3% | | |
| | Above 60 Years | 89 | 7.5% | | |
| Sex of Respondent | Male | 685 | 57.7% | 4.959 | 0.175 |
| | Female | 502 | 42.3% | | |
| Marital Status of the Respondent | Single | 236 | 19.9% | 6.021 | 0.111 |
| | Married | 782 | 65.9% | | |
| | Divorced | 80 | 6.7% | | |
| | Widowed | 89 | 7.5% | | |
| Household size | 2-3 members | 203 | 17.1% | 133.029 | <0.001 |
| | 4-5 members | 333 | 28.1% | | |
| | 6-7 members | 300 | 25.3% | | |
| | 8-9 members | 175 | 14.7% | | |
| | Above 10 members | 176 | 14.8% | | |
| Respondent's Status in household | Household Head | 779 | 65.7% | 17.502 | 0.001 |
| | Spouse of Household Head | 263 | 22.2% | | |
| | Child of Household Head | 97 | 8.2% | | |
| | Dependent | 46 | 3.9% | | |
| Education level of the Respondent | No formal education | 87 | 7.3% | 5.169 | 0.160 |
| | Primary education | 711 | 59.9% | | |
| | Secondary education | 320 | 27.0% | | |
| | Vocational (Skills) | 15 | 1.3% | | |
| | Tertiary education | 54 | 4.5% | | |

Most respondents were married (65.9%), with singles comprising 19.9%, widowed 7.5%, and divorced 6.7%. The analysis showed no significant association between marital status and KAP scores (H = 6.021, $p$ = 0.111), indicating that marital status had minimal impact on anthrax-related awareness or practices.

Household sizes ranged from 2 to over 10 members. The most common household size was 4–5 members (28.1%), followed by 6–7 members (25.3%). Notably, household size was highly significantly associated with KAP scores (H = 133.029, $p$ < 0.001). Larger household sizes may reflect higher levels of interaction and responsibility, thereby influencing knowledge and behavioral practices related to anthrax prevention and control.

Most respondents were household heads (65.7%), with spouses (22.2%), children (8.2%), and dependents (3.9%) making up the rest. This variable was significantly associated with KAP scores (H = 17.502, $p$ = 0.001). This finding suggests that decision-making authority within the household may influence awareness and practices regarding zoonotic diseases like anthrax.

Primary education was the most prevalent (59.9%), followed by secondary education (27.0%), while 7.3% had no formal education. Only 4.5% had tertiary education. Despite this variation, education level was not significantly associated with KAP scores (H=5.169, $p$=0.160). This result suggests that anthrax-related knowledge and practices may be shaped more by experiential or informal knowledge than formal education.

### 3.2. Reliability and internal consistency tests of survey instrument using Kuder-Richardson test (KR-20)

**3.2.1. Reliability and internal consistency test for knowledge-related items.** We assessed respondents' knowledge using nine items. For these knowledge-related items, we obtained a refined KR-20 value of 0.8377 (Table 2), indicating strong internal reliability for KAP related items.

**3.2.2. Reliability and internal consistency test for attitude-related items.** We assessed respondents' attitudes using seven items, yielding a KR-20 value of 0.7290, which also demonstrated acceptable consistency as shown In Table 3.

**Table 2. Refined Kuder-Richardson Formula 20 (KR-20) internal consistency test results retaining 9 Knowledge questions.**

| Knowledge Items | Item difficulty - *proportion of respondents who correctly answer each item* | Item variance – *measure of how much respondents differ in their answers to a particular question* | Item-rest correlation – *correlation between each item and the total test score* |
|---|---|---|---|
| 1. Does Anthrax affect animals? | 0.801 | 0.159 | 0.702 |
| 2. Does Anthrax affect humans? | 0.729 | 0.197 | 0.702 |
| 3. Can acquire Anthrax through contact infected bush meat | 0.662 | 0.224 | 0.604 |
| 4. Can identify an animal that has died from anthrax | 0.370 | 0.233 | 0.506 |
| 5. Vaccination of livestock animals can prevent anthrax | 0.236 | 0.180 | 0.349 |
| 6. Common symptoms of anthrax in humans | 0.694 | 0.212 | 0.474 |
| 7. Symptoms of anthrax in animals | 0.555 | 0.247 | 0.618 |
| 8. Safe to eat meat from an animal that died suddenly | 0.423 | 0.244 | 0.551 |
| 9. Does Anthrax affect animals? | 0.780 | 0.171 | 0.445 |
| **KR20 coefficient is 0.8377** | | | |

**Table 3. Item Analysis of Anthrax Attitude Questions Using Refined Kuder-Richardson Formula 20 (KR-20) for Internal Consistency.**

| Knowledge Items | Item difficulty - *proportion of respondents who correctly answer each item* | Item variance – *measure of how much respondents differ in their answers to a particular question* | Item-rest correlation – *correlation between each item and the total test score* |
|---|---|---|---|
| Believes Anthrax is real | 0.817 | 0.149 | 0.588 |
| How serious is Anthrax infection | 0.708 | 0.207 | 0.577 |
| Would associate with Anthrax survivor | 0.604 | 0.239 | 0.439 |
| How anthrax infection treated in the community | 0.508 | 0.250 | 0.444 |
| Confidence in community hospital system on treating Anthrax | 0.483 | 0.250 | 0.433 |
| **KR20 coefficient is 0.7290** | | | |

### 3.3. Knowledge and attitudes regarding anthrax

A total of 961 respondents (n = 81.0%) responded 'yes' when asked if they had heard about anthrax. Family and friends were a huge source of information for most respondents (48.6%), as well as veterinarians, media (TV, Newspaper, Radio), social media and community leaders. Only 18.1% (n = 215) correctly identified bacteria as the cause of anthrax while the largest proportion (72.1%, n = 856) had no idea. Regarding symptoms in animals and humans, 50.8% and 45.3% did not know respectively. In terms of transmission routes together with possible sources of infection, the majority (63.5%) identified consumption of contaminated meat as a means by which anthrax is transmitted to humans. The belief that vaccination of livestock can prevent anthrax in animals was held by 69.9% (n = 828). A small proportion (4.7%, n = 55) reported household members having suffered from anthrax before, 51.4% were not sure what an anthrax patient looks like, whereas 45.1% identified skin lesions with very few individuals (4.1%) recognizing the respiratory form among others. Alarmingly, 29.1% reported having done nothing when they or a family member fell ill, and 2% either opted to buy medications from a drug store, take themselves or the patient to a traditional healer or treat with home remedies. Only 38.3% visited the nearest health facility.

The overall correct response rate was 58.9%. Only 27.5% (n = 325) were classified as knowledgeable using Bloom's categorization, and 72.5% (n = 860) not knowledgeable. The mean knowledge score was 44.4% (SD ± 21.05108), with a median score of 47.8% as shown in Fig 2A skewness test revealed non-normality in the score distribution (Kurtosis -0.763).

For the attitude scores the mean value was 80.4% (SD ± 12.00723), with a median percentage score of 82.6%. The majority of respondents displayed a Neutral or Positive attitude (81.6%, n = 968) with only 18.4% (n = 219) displaying Poor attitudes.

Most respondents (82.1%, n = 975) believed that anthrax disease exists, and is a very serious disease (71.2%, n = 845), while 3.8% (n = 45) considered it not serious. Regarding the risk of contracting the disease, 66.7% believed they were at risk, while 9.7% (n = 115) were not convinced, and 23.6% (n = 280) were unsure. More than half (60.7%, n = 721) of the respondents said they would relate to a survivor of anthrax, while 21.7% (n = 258). Among those who would not relate with a survivor of anthrax, 90.8% (n = 237) cited fear of contracting the disease, 7.7% (n = 20) mentioned fear of stigma from the community, and 1.5% (n = 4) had other reasons.

**Fig 2. Anthrax Knowledge and Attitude score histograms showing median scores and Bloom's cut-off points.**

When asked about how anthrax is treated in their communities, responses varied from using traditional African medicine-herbal (4.5%, n = 53), spiritual healing (0.9%, n = 10), conventional medicine (54.7%, n = 641), and 40.8% (n = 479) employing other methods. Less than half (48.5%, n = 576) of the respondents expressed high confidence in the community hospital system for treating anthrax, 16.1% (n = 191) said somewhat confident, 12.6% (n = 150) said not confident, and 22.8% (n = 270) said they do not know. Most respondents (62.3%, n = 740) believed it is risky to do business with an anthrax-infected individual, with 18.4% not believing it is risky, and 19.3% not sure. When asked if they would keep information secret if a family member were infected with anthrax, 9.9% (n = 118) said they would, and 8.4% (n = 100) were not sure.

An in-depth analysis revealed statistically significant correlations in attitudes as shown in Table 4.

A Spearman's correlation matrix (Table 5) was generated to assess attitude features against some socio-demographic characteristics. Age was not a statistically significant variable according to Pairwise correlation analyses also revealed significant correlations between knowledge (P < 0.001), attitude (P < 0.001), Practice categories (P = 0.001), and level of education at a 95% confidence level.

### 3.4. Practices

When handling animal carcasses or products, 44.7% (n = 530) reported taking precautions, while 46.3% (n = 550) did not in shown in Table 6. If symptoms of anthrax were noticed, an overwhelming 90.4% (n = 1073) said they would seek medical attention immediately. To ensure the meat they consume is safe, 41.2% (n = 488) buy from trusted sources, while 30.8% (n = 365) take no specific precautions. Additionally, 27.9% (n = 330) mentioned proper cooking, and 21.4% (n = 253) completely avoided wild game meat. Despite concerns about anthrax, only 21.8% (n = 259) had ever reported a suspected case of anthrax or another disease to health authorities, while 78.2% (n = 928) had not. Sudden livestock deaths

**Table 4. Attitudes Toward Anthrax Risk Across Study Districts.**

| Characteristics | Category | District | | | | P. Value |
|---|---|---|---|---|---|---|
| | | Kazungula n (%) | Monze n (%) | Mumbwa n (%) | Mwandi n (%) | < 0.00 |
| Do you believe you have a risk of contracting anthrax disease? (N = 1186) | Yes (n = 791) | 248 (82.) | 288 (75.0) | 182 (45.0) | 73 (75.3) | < 0.00 |
| | No (n = 115) | 25 (8.3) | 33 (8.6) | 50 (12.3) | 7 (7.2) | |
| | Not Sure (n = 280) | 27 (9.0) | 63 (16.4) | 173 (42.7) | 17 (17.5) | |
| Would you relate with a survivor of Anthrax? (N = 1186) | Yes (n = 720) | 196 (65.) | 274 (71.4) | 190 (46.9) | 60 (61.9) | < 0.00 |
| | No (n = 258) | 81 (27.0) | 71 (18.5) | 80 (19.8) | 26 (26.8) | |
| | Not sure (n = 208) | 23 (7.7) | 39 (10.2) | 135 (33.3) | 11 (11.3) | |
| Why would you not relate with a survivor of anthrax? (N = 261) | Fear of contracting the disease (n = 237) | 78 (96.3) | 54 (73.0) | 80 (100) | 25 (96.2) | < 0.00 |
| | Fear of stigma from community (n = 20) | 2 (2.5) | 17 (22.9) | 0 (0) | 1 (3.8) | |
| | Other (n = 4) | 1 (1.2) | 3 (4.1) | 0 (0) | | |
| How confident are you with the community hospital system when it comes to treating anthrax disease? (N = 1186) | I dont know (n = 270) | 45 (15.0) | 61 (15.9) | 145 (35.8) | 19 (19.6) | < 0.00 |
| | Not confident (n = 149) | 47 (15.7) | 22 (5.7) | 64 (15.8) | 16 (16.5) | |
| | Somewhat confident (n = 191) | 29 (9.7) | 73 (19.0) | 57 (14.1) | 32 (32.9) | |
| | Very confident (n = 576) | 179 (59.6) | 228 (59.4) | 139 (34.3) | 30 (31. 0) | |
| Is it risky to do business with someone infected with anthrax disease? (N = 1186) | Yes (n = 740) | 203 (67.7) | 313 (81.5) | 185 (45.7) | 39 (40.2) | < 0.00 |
| | No (n = 217) | 77 (25.7) | 37 (9.6) | 65 (16.0) | 38 (39.2) | |
| | Not Sure (n = 229) | 20 (6.6) | 34 (8.9) | 155 (38.3) | 20 (20.6) | |
| Would you keep information secret if a member of your family were to be infected with Anthrax? (N = 1186) | Yes (n = 117) | 53 (17.7) | 24 (6.3) | 22 (5.4) | 18 (18.6) | < 0.00 |
| | No (n = 969) | 237 (79) | 352 (91.7) | 313 (77.3) | 67 (69.0) | |
| | Not sure (n = 100) | 10 (3.1) | 8 (2.1) | 70 (17.3) | 12 (12.4) | |

**Table 5. Pearson's correlation Matrix on Attitude items.**

| | | Location | Age | Education level | Main source of income |
|---|---|---|---|---|---|
| Do you believe that anthrax disease does exist? | Correlation Coefficient | .278** | -.026 | -.154** | .077** |
| | Sig. (2-tailed) | <0.001 | .365 | <0.001 | .009 |
| | N | 1186 | 1187 | 1187 | 1154 |
| How serious do you think anthrax disease is? | Correlation Coefficient | .272** | -.058* | -.168** | .156** |
| | Sig. (2-tailed) | <0.001 | .045 | <0.001 | <0.001 |
| | N | 1186 | 1187 | 1187 | 1154 |
| Do you believe you have a risk of contracting anthrax disease? | Correlation Coefficient | .270** | .010 | -.147** | .127** |
| | Sig. (2-tailed) | <0.001 | .724 | <0.001 | <0.001 |
| | N | 1186 | 1187 | 1187 | 1154 |
| Would you relate with a survivor of Anthrax? | Correlation Coefficient | .146** | -.010 | -.115** | .203** |
| | Sig. (2-tailed) | <0.001 | .725 | <0.001 | <0.001 |
| | N | 1186 | 1187 | 1187 | 1154 |
| How is anthrax disease treated in your community? | Correlation Coefficient | .075 | -.066 | -.008 | -.044 |
| | Sig. (2-tailed) | .053 | .089 | .837 | .262 |
| | N | 675 | 675 | 675 | 657 |
| How confident are you with the community hospital system when it comes to treating anthrax disease? | Correlation Coefficient | .152** | -.030 | -.059* | .050 |
| | Sig. (2-tailed) | <0.001 | .302 | .041 | .090 |
| | N | 1186 | 1187 | 1187 | 1154 |
| Is it risky to do business (contact) with someone infected with anthrax disease? | Correlation Coefficient | .149** | .035 | -.067* | .050 |
| | Sig. (2-tailed) | <0.001 | .223 | .020 | .091 |
| | N | 1186 | 1187 | 1187 | 1154 |
| Would you keep information secret if a member of your family were to be infected with Anthrax? | Correlation Coefficient | .181** | -.014 | -.079** | .099** |
| | Sig. (2-tailed) | <0.001 | .625 | .007 | .001 |
| | N | 1186 | 1187 | 1187 | 1154 |

(excluding chickens) were reported by 37.0% (n = 438) within these communities in the year before, and the most common action taken was unspecified (44.0%, n = 347), while 23.1% (n = 182) admitted to obtaining meat from the dead animal. Other responses included burying the whole carcass (15.5%, n = 122), burning it (9.0%, n = 71), leaving it to rot in the field (4.3%, n = 34), or reporting it to the area veterinarian.

When asked about sudden deaths of other people's animals in the community, 56.3% (n = 667) were unsure of the frequency, while 16.9% (n = 200) reported it happening once a year. A significant 80.3% (n = 951) believed it was unsafe to cut up carcasses of animals that had died suddenly, yet 8.6% (n = 102) thought it was safe with 19.9% (n = 913) admitted to having obtained hides, bones, or horns from such animals. The most common response regarding the frequency of this practice was uncertainty (39.0%, n = 462), while 9.1% (n = 108) reported doing so once a year. Among respondents, 27.0% (n = 320) reported sudden deaths of game animals in their area, while 40.5% (n = 479) had not. The most common action taken was doing nothing (52.8%, n = 169), while 23.1% (n = 74) reported the deaths to veterinarians. Others either buried (13.4%, n = 43) or burned the whole carcass (4.7%, n = 15), whereas 10.6% (n = 34) admitted to obtaining meat, hides, or other parts from the carcass. On the safety of consuming meat from game animals that died suddenly, 87.3% (n = 1033) believed it was unsafe, while 3.3% (n = 39) thought it was safe. Despite this, 14.9% (n = 176) admitted to having eaten such meat, with 43.2% (n = 76) consuming it once a year.

The mean practice score was 19.3 (SD ± 3.088), and the median score was 19.5. Questions were deliberately structured to assess choices individuals would make. Some individuals consistently selected the risky choice on each question (14.1%). All practice items varied significantly in the four [4] districts (p = 0.000), as shown in Table 6.

**Table 6. Community Practices Related to Anthrax Across Selected Districts in Zambia.**

| Characteristics | Category | District | | | | P. Value |
|---|---|---|---|---|---|---|
| | | Kazungula n (%) | Monzen (%) | Mumbwan (%) | Mwandin (%) | |
| Do you take any precautions when handling animal carcasses that die or animal products?n (N = 1186) | Yes (n = 530) | 158(52.7) | 196(51.0) | 130(32.1) | 46(47.4) | < 0.00 |
| | No (n = 549) | 100(33.3) | 175(45.6) | 238(58.8) | 36(37.1) | |
| | Not Sure (n = 107) | 42(14.0) | 13(3.4) | 37(9.1) | 15(15.5) | |
| If you noticed symptoms of anthrax, would you seek medical attention immediately? (N = 1186) | Yes(n = 1072) | 286(95.3) | 372(96.8) | 326(80.5) | 88(90.7) | < 0.00 |
| | No (n = 37) | 6 (2.0) | 6 (1.6) | 24 (5.9) | 1 (1.03) | |
| | Not Sure(n = 77) | 8 (2.7) | 6 (1.6) | 55 (13.6) | 8 (8.24) | |
| Have you ever reported a suspected case of anthrax or any other disease to health authorities? (N = 1186) | Yes (n = 259) | 79 (26.3) | 101(26.3) | 52(12.8) | 27(27.8) | |
| | No (n = 927) | 221(73.7) | 283(73.7) | 353(87.2) | 70(72.2) | < 0.00 |
| Have any of your animals except chickens died suddenly in the last year? (N = 1183) | Yes(n = 438) | 97 (26.3) | 179(47.0) | 137(33.8) | 25(25.8) | < 0.00 |
| | No (n = 745) | 203 (67.7) | 202(53.0) | 268(66.2) | 72(74.2) | |
| Do you think it is safe to cut up the carcasses of animals that have died suddenly? (N = 1183) | Yes (n = 102) | 26(8.7) | 43 (11.3) | 30 (7.4) | 3 (3.1) | < 0.00 |
| | No (n = 950) | 255 (85.0) | 299(78.5) | 310(76.5) | 86(88.7) | |
| | Not Sure (n = 131) | 19 (6.3) | 39(10.2) | 65 (16.0) | 8 (8.2) | |
| Have you ever gotten hides/bones/horns from an animal that died suddenly (N = 1183) | Yes (n = 236) | 22(7.3) | 160(42.0) | 50 (12.3) | 4 (4.1) | < 0.00 |
| | No (n = 891) | 272 (90.7) | 212(55.6) | 316(78.1) | 91(93.8) | |
| | Not Sure (n = 56) | 6(2.0) | 9(2.4) | 39 (9.6) | 2 (2.1) | |
| Do you think it's safe to get hides/bones/horns from an animal that died suddenly? (N = 1181) | Yes (n = 98) | 26 (8.7) | 45 (11.9) | 22 (5.4) | 5 (5.2) | < 0.00 |
| | No (n = 912) | 261 (87.0) | 263(69.4) | 301(74.3) | 87(89.6) | |
| | Not Sure (n = 171) | 13(4.3) | 71 (18.7) | 82 (20.1) | 5 (5.2) | |
| Have any of the game animals died suddenly in the last 1 year in your area? (N = 1183) | Yes (n = 320) | 148 (49.3) | 47 (12.3) | 87 (21.5) | 38(39.2) | < 0.00 |
| | No (n = 479) | 104 (34.7) | 190 (49.9) | 136 (33.6) | 49 (50.5) | |
| | Not Sure (384) | 48 (16.0) | 144(37.8) | 182(44.9) | 10(10.3) | |
| Do you think it is safe to cut up carcasses of game animals that have died suddenly? (N = 1183) | Yes (n = 46) | 14 (4.7) | 13 (3.4) | 17 (4.2) | 2 (2.1) | < 0.00 |
| | No (n = 1007) | 271 (90.3) | 332(87.1) | 313(77.3) | 91(93.8) | |
| | Not Sure (n = 130) | 15 (5.0) | 36 (9.5) | 75 (18.5) | 4 (4.1) | |
| Do you think it is safe to eat game meat from game animals that died suddenly? (N = 1183) | Yes (n = 39) | 17 (5.7) | 7 (1.8) | 13 (3.2) | 2 (2.1) | < 0.00 |
| | No (n = 1032) | 268 (89.3) | 349(91.6) | 324 (8.0) | 91(93.8) | |
| | Don't know (n = 112) | 15 (5.0) | 25 (6.6) | 68 (16.8) | 4 (4.1) | |
| Have you ever eaten game meat from any game animal that died suddenly? (N = 1183) | Yes (n = 176) | 72 (24.0) | 52 (13.6) | 29 (7.2) | 23(23.7) | < 0.00 |
| | No (n = 1007) | 228 (76.0) | 329(86.4) | 376(92.8) | 74(76.3) | |
| Do you think it's safe to get hides/bones/horns/ scales etc from game animals that died suddenly? | Yes (n = 44) | 9 (3.0) | 21 (5.5) | 9 (2.2) | 5(5.2) | |
| | No (n = 980) | 270 (90.0) | 319(83.7) | 307(75.8) | 84(86.6) | |
| | Don't know (n = 159) | 21 (7.0) | 41 (10.8) | 89 (22.0) | 8 (8.2) | |
| Have you ever gotten hides bones horns scales from game animals that died suddenly? (N = 1183) | Yes (n = 30) | 4 (1.3) | 12 (3.1) | 11 (2.7) | 3 (3.1) | < 0.00 |
| | No (n = 1153) | 296 (98.7) | 369(96.9) | 394(97.3) | 94(96.9) | |

## 3.5. Correlation analysis and Logistic regression modeling

Correlation analysis revealed significant associations between knowledge scores and having heard about the disease ($r = 0.638$, $p < 0.001$). Weaker correlations were observed with prevention and control measures for anthrax ($r = 0.271$, $p < 0.001$), education level ($r = 0.151$, $p < 0.001$), household income ($r = 0.208$, $p < 0.001$), and location ($r = -0.159$, $p < 0.001$).

For example, individuals that had heard about anthrax were more likely to be knowledgeable ($\beta = 0.038$, Exp($\beta$) = 3.268). Additionally, households farther from GMAs were less knowledgeable (Exp($\beta$) > 1), suggesting that proximity to the national park is associated with greater knowledge about the disease. Hearing about anthrax emerged as the strongest predictor of knowledge, while education level and income also positively influenced knowledge. Determinants of poor knowledge using logistic regression are detailed in Table 7. Interaction terms were included to explore potential effect modification, particularly for age, sex, and education. While several interaction terms were not statistically significant, they are reported here for completeness and transparency. Proximity to the national park contributed further to disease awareness. The model's area under the curve was 0.64, with an overall performance of 73.2%, and the Hosmer-Lemeshow goodness-of-fit test indicated a good fit (p = 0.294).

The logistic regression model demonstrated a pseudo-R-squared value of 0.140, and the model was statistically significant ($\chi^2 = 225.126$, p < .001. It had a mean of the dependent variable (0.580, SD = ±0.494), information criteria-Akaike (AIC = 1460.449) and Bayesian (BIC = 1643.270).

Individual attitude scores were correlated to grazing methods practiced (r = 0.122, p = 0.003). Moreover, strong and significant statistical relationships were observed between education, knowledge scores, practices, and attitude scores across the various study locations.

The predictive margins plot (Fig 3) illustrates the probability of holding negative attitudes toward anthrax across revised age categories, stratified by education level. Respondents with no formal education showed the highest predicted probability of negative attitudes, peaking notably in the 25–35 year age group. In contrast, individuals with tertiary education consistently demonstrated the lowest probabilities of negative attitudes across all age groups in comparison to others. Notably, the likelihood of negative attitudes appeared to decline with age especially among those with no formal education. These findings suggest that both age and educational attainment play important roles in shaping attitudes toward anthrax, with younger adults lacking formal education being the most vulnerable group in terms of unfavorable perceptions.

The model demonstrated statistical significance ($\chi^2 = 193.332$, p < 0.001) based on 1,186 observations, with a pseudo-R-squared value of 0.118. The Akaike Information Criterion (AIC) and Bayesian Information Criterion (BIC) were 1521.181 and 1704.001, respectively, indicating model performance. The mean and standard deviation of the dependent variable were 0.519 and 0.500, respectively.

## 4. Discussion

Quantitative findings contained herein endeavored to distinguish between being aware or having heard of and having actual knowledge concerning the topic. Sometimes awareness and knowledge can merge. The study revealed that while a majority (81.0%) of respondents had heard about anthrax, only 27.5% were classified as knowledgeable. These individuals displayed sufficient knowledge of anthrax transmission routes, symptoms, and prevention. The deliberate inclusion of "distinguishing questions," to elicit additional knowledge and provide a clearer differentiation between respondent groups was very useful. Approximately 27.9% believed that proper cooking helps ensure the meat they consume is safe from anthrax. Similar African studies report the existence of a belief that dried or properly cooked meat becomes safe for consumption [24,25]. Some studies [33,26] indicate a link between ingestion of insufficiently cooked meat and anthrax infection. Regarding the signs of anthrax in humans, the cutaneous form was acknowledged by 50.1% of the respondents. Despite the cutaneous form being 95% more prevalent than any other form of anthrax [27] only half of the group recognized it. The gastrointestinal (6.4%), and respiratory forms (5.8%) had very low recognition. Similarly, only 5.8% recognized spore inhalation as a method of contracting the disease. Many of them reported that they sourced information from family and friends which could have attributed to the misconceptions that were perpetuated. Most notably, only 18.1% correctly identified bacteria as the causative agent, while the rest either had no idea or attributed the disease to other factors such as witchcraft or environmental conditions.

**Table 7. Multivariable logistic regression model identifying predictors of poor anthrax knowledge (score <60%) among respondents.**

| Poor knowledge | Coef. | St.Err. | t-value | p-value | [95% Conf | Interval] | Sig |
|---|---|---|---|---|---|---|---|
| Age Group | 1 | . | . | . | . | . | |
| <25 yrs (Ref) | | | | | | | |
| 25-35 yrs | .685 | .918 | -0.28 | .778 | .05 | 9.476 | |
| 35-45 yrs | .231 | .289 | -1.17 | .242 | .02 | 2.681 | |
| 45-55 yrs | .25 | .323 | -1.07 | .283 | .02 | 3.144 | |
| >55 yrs | .13 | .166 | -1.60 | .109 | .011 | 1.574 | |
| Sex of Participants | 1 | . | . | . | . | . | |
| Female (Ref) | | | | | | | |
| Male | .923 | .287 | -0.26 | .798 | .502 | 1.699 | |
| Age and Sex Interaction | 1 | . | . | . | . | . | |
| <25 yrs#Female | .448 | .184 | -1.96 | .05 | .201 | 1.001 | * |
| 25-35 yrs#Male | .697 | .279 | -0.90 | .368 | .318 | 1.528 | |
| 35-45 yrs#Male | 1.167 | .544 | 0.33 | .741 | .467 | 2.912 | |
| 45-55 yrs#Male | .836 | .414 | -0.36 | .717 | .317 | 2.204 | |
| 5. Education level~a | 1 | . | . | . | . | . | |
| No formal education | .144 | .166 | -1.68 | .092 | .015 | 1.375 | * |
| Primary education | .096 | .111 | -2.03 | .042 | .01 | .923 | ** |
| Secondary education | .034 | .045 | -2.51 | .012 | .002 | .476 | ** |
| Age and Education Interaction | 1 | . | . | . | . | . | |
| 25-35 yrs#Primary | 2.411 | 3.202 | 0.66 | .508 | .179 | 32.56 | |
| 25-35 yrs#Secondary | 4.358 | 5.832 | 1.10 | .271 | .316 | 60.038 | |
| 25-35 yrs#Tertiary | 3.422 | 5.308 | 0.79 | .428 | .164 | 71.547 | |
| 35-45 yrs#Primary | 7.183 | 8.93 | 1.59 | .113 | .628 | 82.147 | |
| 35-45 yrs#Secondary | 5.705 | 7.217 | 1.38 | .169 | .478 | 68.08 | |
| 35-45 yrs#Tertiary | 9.044 | 13.531 | 1.47 | .141 | .482 | 169.768 | |
| 45-55 yrs#Primary | 5.596 | 7.169 | 1.34 | .179 | .454 | 68.937 | |
| 45-55 yrs#Secondary | 3.987 | 5.281 | 1.04 | .296 | .297 | 53.473 | |
| 45-55 yrs#Tertiary | 6.978 | 11.45 | 1.18 | .236 | .28 | 173.98 | |
| >55 yrs#Primary | 15.413 | 19.611 | 2.15 | .032 | 1.273 | 186.593 | ** |
| >55 yrs#Secondary | 18.224 | 24.095 | 2.20 | .028 | 1.365 | 243.261 | ** |
| >55 yrs#Tertiary | 32.563 | 54.916 | 2.07 | .039 | 1.195 | 887.594 | ** |
| **HLW Interface** | | | | | | | |
| Blue Lagoon National Park | 1.848 | .411 | 2.76 | .006 | 1.195 | 2.857 | *** |
| Simalaha Conservancy | .324 | .069 | -5.30 | 0 | .213 | .491 | *** |
| **Keep Mammalian Livestock** | | | | | | | |
| Chickens Only (Ref) | 1 | . | . | . | . | . | |
| Mammals | .692 | .106 | -2.41 | .016 | .513 | .933 | ** |
| **Income level** | **1** | . | . | . | . | . | |
| <K1100 (Ref) | | | | | | | |
| K1100-2000 | .624 | .104 | -2.83 | .005 | .45 | .865 | *** |
| K2100-3000 | .517 | .126 | -2.71 | .007 | .321 | .832 | *** |
| K3100-4000 | .247 | .09 | -3.82 | 0 | .12 | .506 | *** |
| K4100-5000 | .513 | .201 | -1.70 | .088 | .238 | 1.106 | * |
| Above K5000 | .433 | .14 | -2.59 | .01 | .23 | .815 | *** |
| Distance from National Park | 1 | . | . | . | . | . | |

*(Continued)*

**Table 7.** (Continued)

| Poor knowledge | Coef. | St.Err. | t-value | p-value | [95% Conf | Interval] | Sig |
|---|---|---|---|---|---|---|---|
| <5km | | | | | | | |
| 5-10km | 1.533 | .393 | 1.67 | .096 | .928 | 2.533 | * |
| 10-25km | .695 | .147 | -1.72 | .086 | .458 | 1.053 | * |
| >25 | .615 | .142 | -2.10 | .036 | .391 | .968 | ** |

Key: *p < 0.05, **p < 0.01, **p < 0.001.

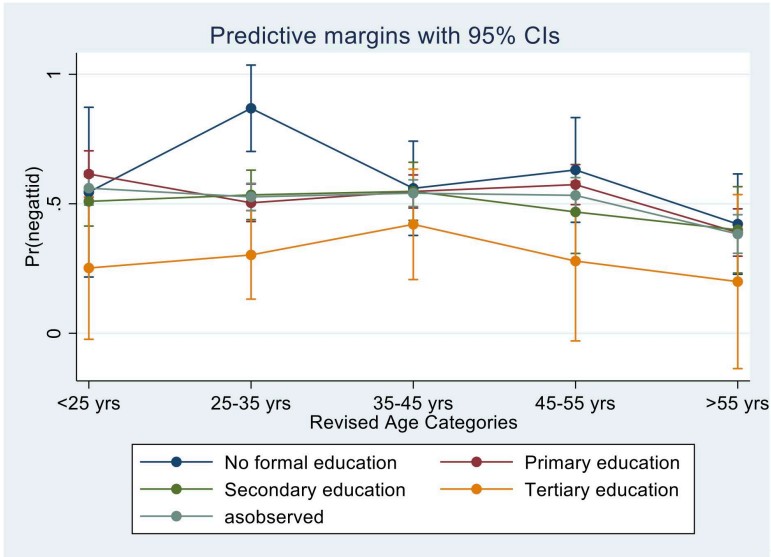

**Fig 3. Predictive Margin plot indicating the impact of education on negative attitudes across age groups.**

The findings reveal significant knowledge gaps and misconceptions about anthrax, despite widespread awareness. Misconceptions, such as believing properly cooked meat is safe indicate misinformation likely spread through informal sources. Limited recognition of anthrax forms, as well as modes of transmission, suggests a lack of understanding of its severity and directly translates into practices. These gaps highlight the need for targeted education campaigns, emphasizing accurate disease information, proper prevention measures, and the risks associated with unsafe meat consumption practices.

Furthermore, these findings align with studies in Zambia and other sub-Saharan African countries, which highlight knowledge deficiencies as key barriers to effective zoonotic disease control [8,24]. The low awareness levels may stem from limited veterinary extension services, inadequate health education, and reliance on informal sources of information, such as family and community elders.

There were positive and significant correlations between knowledge-practice (r = 132, p=<0.001), knowledge-attitudes (r = 196, p=<0.001), and attitude-practice (r = 163, p=<0.001). These findings were consistent with findings by Al Ahdab [28]. This proves that the lack of formal education influenced knowledge categorization. From this study, only 32.8% attained secondary education. Education influences one's access to information and ability to comprehend health promotion messages [24]. Despite these knowledge gaps, 69.9% of respondents believed that vaccination of livestock could prevent anthrax, indicating a level of awareness regarding preventive measures. However, this belief was not strongly

linked to proactive vaccination behavior, as many participants lacked access to veterinary services or did not perceive vaccination as a priority. This disconnect between knowledge and action has been observed in similar studies, where even individuals aware of disease prevention strategies fail to adopt protective measures due to economic, logistical, and cultural constraints [6,9].

The study revealed several determinants of poor knowledge about anthrax among participants aged over 55 with primary education (OR = 15.413, p = .032), secondary education (OR = 18.224, p = .028), and tertiary education (OR = 32.563, p = .039) having significantly higher likelihoods of poor knowledge about anthrax. Those associated with Blue Lagoon National Park (OR = 1.848, p = .006) also had significantly higher odds of poor knowledge about anthrax mainly because most areas around have never had any case of anthrax disease. In contrast, participants with no formal education (OR = 0.144, p = 0.092), primary education (OR = 0.096, p = 0.042), and secondary education (OR = 0.034, p = 0.012) exhibited significantly lower likelihoods of poor knowledge about anthrax.

Additionally, participants who kept mammals (OR = 0.692, p = .016) had a significantly lower likelihood of poor knowledge about anthrax. Income levels were also a significant determinant, with participants earning between K1100-2000 (OR = 0.624, p = .005), K2100-3000 (OR = 0.517, p = .007), K3100-4000 (OR = 0.247, p = < 0.001), and above K5000 (OR = 0.433, p = .010) significantly less likely to have poor knowledge about anthrax.

Furthermore, participants living more than 25 km from the national park (OR = 0.615, p = .036) exhibited significantly lower odds of poor knowledge about anthrax. Any contrast in awareness and knowledge suggests that there is a need for targeted education or interventions.

Attitudinal analysis revealed that 82.1% of respondents believed anthrax was a serious disease, yet only 66.7% perceived themselves as being personally at risk. This discrepancy between general and personal risk perception is concerning, as it may influence compliance with disease prevention measures. Similar patterns have been reported in studies on zoonotic diseases, where individuals acknowledge severity of a disease but underestimate their vulnerability due to optimistic bias or cultural normalization of risk behaviors [29].

Community stigma around anthrax survivors was another key finding. While 60.7% reported they would associate with a survivor, 21.7% stated they would avoid them, citing fear of contagion. This stigma could discourage infected individuals from seeking timely medical care or disclosing their condition, further complicating disease control efforts. Furthermore, a significant proportion of respondents (40.8%) still believed in traditional healing methods, including herbal remedies and spiritual interventions, for treating anthrax. These findings underscore the need for integrated public health messaging that respects local beliefs while promoting scientifically proven treatment options [3].

Risk behaviors related to anthrax exposure were prevalent in the study areas. While 87.3% of respondents recognized that consuming meat from animals that died suddenly was unsafe, 14.9% admitted to having done so, primarily due to food insecurity. Additionally, 19.9% reported handling carcasses or obtaining animal by-products (hides, bones, horns) from dead animals, despite being aware of the risks. These findings suggest that economic pressures often outweigh health concerns, leading to continued engagement in high-risk activities. A study in Zimbabwe highlighted similar challenges, such as limited community knowledge about anthrax and risky practices like leaving carcasses to decompose in the environment. This study also emphasized the need for multidisciplinary efforts to improve surveillance and control measures [10].

The illegal game meat trade emerged as a significant pathway for anthrax transmission. Similar studies in Zambia and other regions of Africa have found that bushmeat consumption is deeply embedded in local culture and economic survival strategies, making outright bans difficult to enforce [18,22]. Instead, interventions should focus on promoting safer hunting and meat-handling practices, coupled with livelihood diversification programs to reduce economic dependence on wildlife trade.

Socioeconomic constraints played a significant role in shaping community behaviors toward anthrax prevention. The study found that 52.6% of respondents had household incomes below ZMW 1,000 ($35), with many engaging in

subsistence farming and informal employment. Limited financial resources restrict access to veterinary services, proper meat inspection, and alternative protein sources, increasing reliance on high-risk food sources such as bushmeat.

Furthermore, only 29.5% of respondents benefited from social cash transfer programs, and many received little to no external assistance during periods of drought or food shortages. These economic hardships drive behaviors that facilitate anthrax transmission, highlighting the need for integrated One Health approaches that address both public health and economic resilience. Strengthening community-based surveillance, expanding social protection programs, and improving access to veterinary services could significantly enhance disease prevention efforts.

In Zambia, a study on the socio-economic determinants of anthrax transmission found that poverty, cultural practices, and limited access to veterinary services were key drivers of risky behaviors, such as consuming meat from animals that died suddenly. These findings align with the current study's emphasis on the role of socio-economic factors [3,30]. Another Zambian study explored lay perceptions and cultural practices influencing anthrax transmission in cattle. It found that mistrust of veterinary services and traditional practices, such as exchanging animals between herds, contributed to disease persistence [24] which is agreement with our findings.

## 4.1. Implications for policy and disease control strategies

The findings of this study have several implications for anthrax control, public health interventions, and wildlife conservation policies in Zambia:

i. Enhanced Community Education – Targeted risk communication campaigns should be developed to improve anthrax awareness, correct misinformation, and promote safer practices. Leveraging local influencers, community leaders, and social media platforms could enhance outreach effectiveness.

ii. Strengthening Veterinary Services – Expanding livestock vaccination programs, improving disease reporting mechanisms, and ensuring timely access to veterinary care are critical for reducing anthrax outbreaks in livestock and minimizing spillover to humans.

iii. Regulation of Wildlife Trade – Policy efforts should focus on monitoring and regulating the game meat value chain, identifying high-risk nodes in illegal wildlife trade, and promoting alternative livelihoods to reduce economic dependence on bushmeat.

iv. Integration of One Health Approaches – Strengthening cross-sectoral collaboration between veterinary, public health, and conservation agencies will be essential for effective anthrax surveillance, response, and prevention.

## 4.2. Study limitations and future research

While this study provides valuable insights into the Knowledge, Attitudes, and Practices (KAPs) of communities regarding anthrax transmission and wildlife trade value chains in Zambia, several limitations should be acknowledged:

*Self-Reported Data and Social Desirability Bias* - The study relied heavily on self-reported responses from surveys. Participants may have underreported or misrepresented their behaviors, particularly regarding illegal wildlife trade, bushmeat consumption, and risky handling of animal carcasses, due to fear of legal repercussions or social stigma. This social desirability bias could have led to an underestimation of high-risk behaviors. Future research could incorporate direct observations, ethnographic methods, or forensic investigations of wildlife trade networks to validate findings.

*Recall Bias* - Some aspects of the study required participants to recall past experiences, such as anthrax outbreaks, livestock losses, or personal encounters with the disease. Given that many events may have occurred months or even years before the survey, recall bias could have affected the accuracy of responses. Future studies could mitigate this limitation by utilizing longitudinal study designs or real-time disease reporting mechanisms to capture more precise data.

*Limited Generalizability* - The study was conducted in Simalaha Conservancy, Blue Lagoon, and Lochinvar National Parks, which are geographically and socio-culturally distinct from other regions in Zambia. The findings may not be fully generalizable to other game management areas (GMAs), national parks, or urban communities where wildlife trade and anthrax exposure risks may differ. Future research should consider expanding the study to multiple ecological zones and economic settings to enhance the representativeness of findings.

*Challenges in Verifying Illegal Wildlife Trade Data* - Due to the sensitive nature of illegal game meat trade, some participants may have been unwilling to disclose full details about their involvement. While qualitative methods such as key informant interviews (KIs) and FGDs helped gather contextual insights, triangulation with law enforcement records, wildlife seizure reports, or DNA barcoding of bushmeat could provide stronger validation of illegal wildlife trade networks.

*Sample Size and Response Rates* - While the study only targeted 1,155 respondents, and got more respondents, missing data on the geographical setup, some survey responses were incomplete or had to be excluded due to inconsistencies or missing data. Future studies could increase sample diversity and use stratified random sampling to improve representativeness.

*Ethical and Cultural Sensitivities* - Discussions about anthrax outbreaks, livestock losses, and bushmeat consumption touched on culturally sensitive and livelihood-related topics. Some participants may have been hesitant to share information due to superstitions, traditional beliefs, or mistrust of researchers. While community engagement strategies were used to build trust and ensure ethical research practices, deeper participatory approaches such as co-designing interventions or closed ballot rankings or responses during discussions with local communities could enhance openness and data accuracy in future studies.

*Cross-Sectional Study Design* - This study employed a cross-sectional design, capturing data at a single point in time. While this approach provided a snapshot of community KAPs, it did not assess changes over time. Future studies should incorporate longitudinal designs to track how knowledge, attitudes, and practices evolve, particularly in response to interventions or changes in disease dynamics.

*Limited Integration of Environmental and Ecological Data* - Although the study examined human behaviors, it did not extensively analyze environmental factors influencing anthrax persistence, such as soil characteristics, water sources, and climate variability. Integrating geospatial mapping, ecological modeling, and environmental sampling in future research could provide a more comprehensive understanding of anthrax transmission dynamics at the human-wildlife-livestock interface.

## 5. Conclusions and recommendations

This study examined Knowledge, Attitudes, and Practices (KAPs) related to anthrax transmission and wildlife trade value chains among communities living in and around Simalaha Conservancy, Blue Lagoon, and Lochinvar National Parks in Zambia. The findings reveal that while anthrax is widely recognized as a serious zoonotic disease, significant knowledge gaps, misconceptions, and risky practices persist, contributing to its continued transmission. Socioeconomic factors, including food insecurity, limited veterinary services, and reliance on illegal wildlife trade, further exacerbate the risks associated with anthrax outbreaks.

Despite a high level of awareness (81%) about anthrax, only 27.5% of respondents demonstrated sufficient knowledge regarding its transmission, symptoms, and preventive measures. Misinformation, such as beliefs in traditional healing and misconceptions about anthrax sources, undermines efforts to control the disease. Attitudes towards anthrax varied, with stigma against survivors and skepticism towards formal healthcare affecting disease response and management. Additionally, while 87.3% acknowledged the dangers of consuming meat from animals that died suddenly, 14.9% admitted to doing so, highlighting the role of economic hardship in driving unsafe food practices.

The illegal wildlife trade emerged as a key factor in anthrax transmission, driven by economic necessity and cultural preferences for bushmeat. However, weak enforcement of wildlife trade regulations and limited alternative livelihood opportunities sustain the demand for illegally sourced game meat, further increasing anthrax exposure risks.

The study highlights the urgent need for targeted interventions, including enhanced public health education, improved veterinary services, strengthened wildlife trade regulations, and alternative livelihood programs to reduce community dependence on risky behaviors.

## Acknowledgments

The authors are grateful to staff members of the Department of Veterinary Services in the Ministry of Livestock and Fisheries in Monze, Kazungula and Mumba District for facilitating data collection. AI was used for qualitative data analysis and to improve the language of this article: OpenAI. (2024). ChatGPT [Large language model]. https://chatgpt.com.

## Author contributions

**Conceptualization:** Chitwambi Makungu, Mtui-Malamsha N. Jesse, Suwilanji S. Sichone, Mwila Kayula, Geoffrey Mainda, Fredrick M. Kivaria, Charles Bebay, Baba Soumare, Chisoni Mumba.

**Data curation:** Exillia Kabbudula, Laila Gondwe, Chitwambi Makungu, Kezzy Besa, Chisoni Mumba.

**Formal analysis:** Exillia Kabbudula, Laila Gondwe, Chitwambi Makungu, Kezzy Besa, Noanga Mebelo, Chisoni Mumba.

**Funding acquisition:** Chitwambi Makungu.

**Investigation:** Exillia Kabbudula, Laila Gondwe, Chitwambi Makungu, Suwilanji S. Sichone, Chisoni Mumba.

**Methodology:** Exillia Kabbudula, Laila Gondwe, Chitwambi Makungu, Chisoni Mumba.

**Project administration:** Chitwambi Makungu, Mtui-Malamsha N. Jesse, Chisoni Mumba.

**Resources:** Chitwambi Makungu, Mtui-Malamsha N. Jesse, Suze P. Filippini, Chisoni Mumba.

**Software:** Exillia Kabbudula, Laila Gondwe, Chitwambi Makungu, Chisoni Mumba.

**Supervision:** Chisoni Mumba.

**Validation:** Exillia Kabbudula, Laila Gondwe, Chitwambi Makungu, Mtui-Malamsha N. Jesse, Kezzy Besa, Suwilanji S. Sichone, Noanga Mebelo, Mwila Kayula, Geoffrey Mainda, Fredrick M. Kivaria, Charles Bebay, Baba Soumare, Suze P. Filippini, Chisoni Mumba.

**Visualization:** Exillia Kabbudula, Laila Gondwe, Chitwambi Makungu, Mtui-Malamsha N. Jesse, Kezzy Besa, Suwilanji S. Sichone, Noanga Mebelo, Mwila Kayula, Geoffrey Mainda, Fredrick M. Kivaria, Charles Bebay, Baba Soumare, Suze P. Filippini, Chisoni Mumba.

**Writing – original draft:** Exillia Kabbudula, Laila Gondwe, Chitwambi Makungu, Mtui-Malamsha N. Jesse, Kezzy Besa, Suwilanji S. Sichone, Noanga Mebelo, Mwila Kayula, Geoffrey Mainda, Fredrick M. Kivaria, Charles Bebay, Baba Soumare, Suze P. Filippini, Chisoni Mumba.

**Writing – review & editing:** Exillia Kabbudula, Laila Gondwe, Chitwambi Makungu, Mtui-Malamsha N. Jesse, Kezzy Besa, Suwilanji S. Sichone, Noanga Mebelo, Mwila Kayula, Geoffrey Mainda, Fredrick M. Kivaria, Charles Bebay, Baba Soumare, Suze P. Filippini, Chisoni Mumba.

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
