## [Decision Letter · Decision Letter 0]

30 Jun 2025

Knowledge, Attitudes, and Practices on Anthrax in Selected Game Management Areas in Zambia

Dear Dr. Mumba,

Thank you for submitting your manuscript to PLOS Neglected Tropical Diseases. After careful consideration, we feel that it has merit but does not fully meet PLOS Neglected Tropical Diseases's publication criteria as it currently stands. Therefore, we invite you to submit a revised version of the manuscript that addresses the points raised during the review process.

Please submit your revised manuscript within 60 days Aug 29 2025 11:59PM. If you will need more time than this to complete your revisions, please reply to this message or contact the journal office at plosntds@plos.org. Please include the following items when submitting your revised manuscript:

We look forward to receiving your revised manuscript.

Kind regards,

Jeff Gilbert

Guest Editor

Elsio Wunder Jr

Section Editor

Shaden Kamhawi

co-Editor-in-Chief

Paul Brindley

co-Editor-in-Chief

**Additional Editor Comments (if provided):**

I concur with all recommendations and changes provided by the 3 reviewers. With one requiring major revision for acceptance the overall decision is thus major revision. We look forward to receiving an updated manuscript.

**Journal Requirements:**

**Reviewers' Comments:**

Reviewer's Responses to Questions

**Key Review Criteria Required for Acceptance?**

**Methods**

-Are the objectives of the study clearly articulated with a clear testable hypothesis stated?

-Is the study design appropriate to address the stated objectives?

-Is the population clearly described and appropriate for the hypothesis being tested?

-Is the sample size sufficient to ensure adequate power to address the hypothesis being tested?

-Were correct statistical analysis used to support conclusions?

-Are there concerns about ethical or regulatory requirements being met?

Reviewer #1: The methods were well described with details on the study location, design and sampling. A correction should be done on line 146 to 147 where it is stated the sample size was 1,155 and after removing the incomplete questionnaires remained with sample size of 1,185 which is higher?

Reviewer #2: Methods: While I commend the authors of this study for acquiring such a large sample size, the methods section needs to be strengthened

-Rationale of the sample size formula and parameters used (including the estimated proportion of 50% and the design effect of 3). This needs to be clearly stated as part of your methods.

Why the authors decided not to use a sample size formula that accounts for clustering?

While it is implied through the methods, the unit of analysis needs to be clearly stated.

There is no description on how the sampling and recruitment was done, this is crucial as this can impact results. How was clustering taken into consideration when identifying potential participants?

Analysis: This section is lacking information and needs further work. There is no information on:

Different analysis done and rationale behind it

The model-building process, clearly state the outcome, what variables were considered for inclusion, and significance cut-off determined.

Reviewer #3: - The described objectives and methods are consistent.

- Did you collaborate with anthropologists for the development of the questionnaires? If so, it would be good to mention it in the methods section.

- To strengthen the methodology, I recommend adding explanations about the use of logistic regression in the Data Management and Analysis section

- Were all the surveys administered online? If that is the case, I suggest considering this aspect in the 'Study Limitations' section. Because you cannot be certain that the study population meets the inclusion criteria defined in the 'Sampling Strategy' section.

**Results**

-Does the analysis presented match the analysis plan?

-Are the results clearly and completely presented?

-Are the figures (Tables, Images) of sufficient quality for clarity?

Reviewer #1: Results were well described and presented in appropriate tables

Reviewer #2: There are some sections under the results section that are lacking clarity, the results do not necessarily match the analysis plan described as there was very little information provided under the methods section.

In particular: It seems like the authors do not describe the logistic regression model results in a succinct way. The authors present part of the results (not discussed under the results section) as part of the discussion.

The authors should present the socio demographic data by site and include additional information on completeness of data, as this will assist the reader in understanding the impact of the conclusions.

Table 8:

The title needs to be edited to further explain the model and results.

How was poor knowledge determined based on what criteria? This needs to be part of the model building description as part of the analysis.

Where all these variables part of the final model? It is hard to think that find significance with so many variables (and categories within each variable). This needs to be addressed as part of the methods (see comment above)

There appears to be missing some information like footnotes to explain what the asterisks mean (this is applicable to table 5)

The authors should add a description of Figure three as it would be difficult for the reader to understand what the authors are presenting.

Reviewer #3: - The results are clearly presented, and the study is highly interesting

- Some results appear to be included in the Methods section, which may need to be revised. I have indicated them directly in the document.

**Conclusions**

-Are the conclusions supported by the data presented?

-Are the limitations of analysis clearly described?

-Do the authors discuss how these data can be helpful to advance our understanding of the topic under study?

-Is public health relevance addressed?

Reviewer #1: Conclusion is based on the findings. However, for lines 653 to 666, please remove results data from the statements to keep the simple.

Reviewer #2: Public health relevance of the topic is addressed, and the authors do compare and contrast with other studies performed in the region. The limitations were clearly described.

One conclusion point the authors make that I don't agree is supported by the study is that according to the authors, the illegal wildlife trade emerged as a key factor in anthrax transmission. Anthrax outbreaks due to spillover are unlikely to be associated with wildlife trade. Historically, human anthrax follows wildlife die-offs due to people butchering the meat and selling it for consumption once they find the carcass.

Reviewer #3: The conclusion is clearly and effectively written.

**Editorial and Data Presentation Modifications?**

Reviewer #1: Recommended Minor revision

Reviewer #2: Tables and Figure will benefit from descriptive titles that will aid the reader to understand the data being presented. In addition, many of the tables lack foot notes describing superscripts used.

Reviewer #3: (No Response)

**Summary and General Comments**

Reviewer #1: I find the paper well structured and written apart from the minor corrections indicated.

Reviewer #2: While I believe this is interesting work, there authors should expand on methods and results so the reader can understand the significance and impact of this study. Detail information on analysis and missing data is lacking, which will have an impact in understanding results and the broader implications of the conclusions. The Authors are trying to link anthrax to illegal wildlife trades and conservation threats. While I agree it is a threat from a conservation standpoint, it is not necessarily the case in trade situations. Anthrax is not spread between live animals, and most cases we observe (at least those recorded) are more directly associated to the butchering and consumption of a carcass that was found.

Reviewer #3: This is a very interesting article. The methods are rigorous, and the results are well presented. Just a few adjustments regarding the different sections: the 'Sampling Strategy' section should be placed before the 'Description of the Study Sites' section. Some results appear in the Methods section and should be moved to the Results section."

The sections I recommend moving are indicated directly in the document.

PLOS authors have the option to publish the peer review history of their article (what does this mean?). If published, this will include your full peer review and any attached files.

Reviewer #1: **Yes: ** Lawrence Mugisha

Reviewer #2: No

Reviewer #3: **Yes: ** Nivohanitra Perle RAZAFINDRAIBE

**Figure resubmission:**

**Reproducibility:**



---

## [Decision Letter · Decision Letter 1]

24 Sep 2025

Dear Dr Mumba,

We are pleased to inform you that your manuscript 'Knowledge, Attitudes, and Practices on Anthrax in Selected Game Management Areas in Zambia' has been provisionally accepted for publication in PLOS Neglected Tropical Diseases.

Best regards,

Jeff Gilbert

Guest Editor

Elsio Wunder Jr

Section Editor

Shaden Kamhawi

co-Editor-in-Chief

Paul Brindley

co-Editor-in-Chief

Reviewer #1:

Reviewer #2:

Reviewer's Responses to Questions

**Key Review Criteria Required for Acceptance?**

**Methods**

-Are the objectives of the study clearly articulated with a clear testable hypothesis stated?

-Is the study design appropriate to address the stated objectives?

-Is the population clearly described and appropriate for the hypothesis being tested?

-Is the sample size sufficient to ensure adequate power to address the hypothesis being tested?

-Were correct statistical analysis used to support conclusions?

-Are there concerns about ethical or regulatory requirements being met?

Reviewer #1: All reviewer comments addressed regarding method section

Reviewer #2: The authors addressed the reviewers concerns and recommendations.

**Results**

-Does the analysis presented match the analysis plan?

-Are the results clearly and completely presented?

-Are the figures (Tables, Images) of sufficient quality for clarity?

Reviewer #1: Results clearly and completely presented

Reviewer #2: The authors addressed the reviewers concerns and recommendations.

**Conclusions**

-Are the conclusions supported by the data presented?

-Are the limitations of analysis clearly described?

-Do the authors discuss how these data can be helpful to advance our understanding of the topic under study?

-Is public health relevance addressed?

Reviewer #1: Conclusions okay

Reviewer #2: The authors addressed the reviewers concerns and recommendations.

**Editorial and Data Presentation Modifications?**

Reviewer #1: N/A

Reviewer #2: The authors addressed the reviewers concerns and recommendations.

**Summary and General Comments**

Reviewer #1: N/A based on previous comments

Reviewer #2: The authors addressed the reviewers concerns and recommendations.

PLOS authors have the option to publish the peer review history of their article (what does this mean?). If published, this will include your full peer review and any attached files.

Reviewer #1: **Yes: ** Lawrence Mugisha

Reviewer #2: No

---

## [Editor Report · Acceptance letter]

Dear Dr Mumba,

We are delighted to inform you that your manuscript, "Knowledge, Attitudes, and Practices on Anthrax in Selected Game Management Areas in Zambia," has been formally accepted for publication in PLOS Neglected Tropical Diseases.

Best regards,

Shaden Kamhawi

co-Editor-in-Chief

Paul Brindley

co-Editor-in-Chief
